# Oral Administration of Probiotic *Bifidobacterium breve* Ameliorates Tonic–Clonic Seizure in a Pentylenetetrazole-Induced Kindling Mouse Model via Integrin-Linked Kinase Signaling

**DOI:** 10.3390/ijms25179259

**Published:** 2024-08-27

**Authors:** Toshiaki Ishii, Motohiro Kaya, Yoshikage Muroi

**Affiliations:** 1Department of Basic Veterinary Medicine, Obihiro University of Agriculture and Veterinary Medicine, Obihiro 080-8555, Hokkaido, Japan; muroi@obihiro.ac.jp; 2Center for Industry-University Collaboration, Obihiro University of Agriculture and Veterinary Medicine, Obihiro 080-8555, Hokkaido, Japan; mkaya@obihiro.ac.jp

**Keywords:** epilepsy, pentylenetetrazole (PTZ), *Bifidobacterium breve* A1 (*B. breve* A1), seizure, integrin-linked kinase (ILK), integrin, Cpd22, neuropsin, Akt, MMP-9, hippocampus

## Abstract

Epilepsy is a chronic neurological disorder characterized by recurrent seizures that affects over 70 million people worldwide. Although many antiepileptic drugs that block seizures are available, they have little effect on preventing and curing epilepsy, and their side effects sometimes lead to serious morbidity. Therefore, prophylactic agents with anticonvulsant properties and no adverse effects need to be identified. Recent studies on probiotic administration have reported a variety of beneficial effects on the central nervous system via the microbiota–gut–brain axis. In this study, we investigated the effects of the oral administration of *Bifidobacterium breve* strain A1 [MCC1274] (*B. breve* A1) on tonic–clonic seizure in a pentylenetetrazole (PTZ)-induced kindling mouse (KD mouse) model. We found that the oral administration of *B. breve* A1 every other day for 15 days significantly reduced the seizure score, which gradually increased with repetitive injections of PTZ in KD mice. The administration of *B. breve* A1, but not saline, to KD mice significantly increased the level of Akt Ser^473^ phosphorylation (p-Akt) in the hippocampus; this increase was maintained for a minimum of 24 h after PTZ administration. Treatment of *B. breve* A1-administered KD mice with the selective inhibitor of integrin-linked kinase (ILK) Cpd22 significantly increased the seizure score and blocked the antiepileptic effect of *B. breve* A1. Moreover, Cpd22 blocked the *B. breve* A1-induced increase in hippocampal p-Akt levels. These results suggest that the ILK-induced phosphorylation of Akt Ser^473^ in the hippocampus might be involved in the antiepileptic effect of *B. breve* A1.

## 1. Introduction

Epilepsy is a chronic neurological disorder characterized by irregular and recurrent seizures that affects over 70 million people worldwide [1]. Although numerous antiepileptic drugs that block seizures are available, they have little effect on cures for epilepsy and do not adequately support all affected patients [2,3]. As epilepsy treatment continues essentially throughout life and the side effects of the antiepileptic drugs currently available sometimes lead to serious morbidity [4], it is necessary to identify prophylactic agents with anticonvulsant properties but no adverse effects.

It recently became widely known that probiotics provide beneficial effects on human and animal health through the daily intake of adequate amounts of living microorganisms [5]. Above all, the application of some probiotics has been reported to be beneficial for the central nervous system via the microbiota–gut–brain axis [6,7,8]. Recently, the efficacy of fecal microbiota transplantation in preventing seizure relapse after withdrawing antiepileptic drugs has been confirmed, suggesting the possibility of a novel treatment for epilepsy through the remodeling of the gut microbiota [9]. Moreover, an epilepsy-linked gut microbiota signature in a pediatric rat model of acquired epilepsy has been reported [10]. Therefore, we expect that some probiotics that have not been examined in terms of their antiepileptic effects might be available as treatment for epilepsy in the future.

We previously reported that oral administration of *Bifidobacterium breve* strain A1 [MCC1274] (*B. breve* A1) leads to the restoration of abnormal hippocampal synaptic plasticity and cognitive deficits in Parkinson’s disease model mice (PD mice) [11]. *B. breve* A1 is also known to prevent cognitive impairment in Alzheimer’s disease (AD) model mice [12] and patients with AD [13]. On the other hand, cognitive impairment is often seen as secondary to or caused by epilepsy and is considered a common comorbidity of epilepsy along with mood and behavioral problems in patients with epilepsy [14]. Therefore, epilepsy and cognitive impairment may be connected to each other bidirectionally in the pathological process. Because *B. breve* A1 restores abnormal hippocampal synaptic plasticity and cognitive deficits [11], its potential pharmaceutical use for epilepsy prevention is expected.

Among the available animal models of seizures, the chronic pentylenetetrazole (PTZ)-induced kindling model mice (KD mice) produced by the repeated administration of PTZ at a subconvulsive dose show a more similar pathology to that of human epilepsy than do acute seizure models [15]. In addition, KD mice show intensified seizure activity and/or enhanced seizure susceptibility severity in a stepwise manner with each PTZ injection [15,16,17]. Mizoguchi et al. [18] found that repeated treatment with PTZ increased both the activity and protein expression levels of hippocampal matrix metalloproteinase 9 (MMP-9), which promotes neuronal plasticity through the cleavage of extracellular matrix (ECM) proteins [19,20], accompanied by kindled seizures. Moreover, activated MMP-9 not only modulates hippocampal synaptic physiology through integrin receptors [20] but also contributes to integrin signaling disruption and subsequent apoptotic cell death in the hippocampus after pilocarpine- and kainite-induced seizures [21,22]. In contrast, treatment with a selective MMP-9 inhibitor has been reported to attenuate MMP-9 upregulation, β1-integrin degradation, the reduction in integrin-mediated survival signaling, and subsequent hippocampal damage after pilocarpine-induced seizures [21].

In the present study, to evaluate the antiepileptic potential of *B. breve* A1, we investigated the effects of oral administration of *B. breve* A1 on tonic–clonic seizure in PTZ-induced KD mice. We also examined the effects of *B. breve* A1 administration on the hippocampal mRNA and protein expression levels of several proteins known to be involved in synaptic plasticity and reported to change in epilepsy animal models.

## 2. Results

### 2.1. Administration of B. breve A1 to KD Mice Ameliorated PTZ-Induced Seizures

We examined the effects of oral administration of *B. breve* A1 on PTZ-induced seizures in KD mice. PTZ (37 mg/kg) was administered to KD mice intraperitoneally every other day for 15 days (Figure 1A). Figure 2 shows that the mean seizure score gradually increased with repeated injections of PTZ. Oral administration of *B. breve* A1 to KD mice every other day partially but significantly reduced the mean seizure score after the fourth PTZ injection compared with saline-administered KD mice (Figure 2). No significant differences in the seizure score after the first to the third administrations of PTZ were found between the saline- and *B. breve* A1-administered KD mice. On the other hand, oral administration of nonviable *B. breve* A1 did not have any significant effects on the mean seizure score after PTZ injection in KD mice (Figure 3).

### 2.2. Effect of Administration of B. breve A1 on the Hippocampal mRNA Expression Levels of Neuropsin, MMP-9, Integrin-Linked Kinase (ILK), Brain-Derived Neurotrophic Factor (BDNF), and Ionized Calcium-Binding Adapter Molecule 1 (Iba1)

As epileptogenicity is considered to be associated with changes in synaptic plasticity [23,24,25], we examined the mRNA expression levels of several proteins known to be involved in synaptic plasticity and reported to change in epilepsy animal models. The mRNA expression levels of neuropsin, a plasticity-related extracellular protease [26], and ILK, an integrin-dependent neuroprotective signaling molecule [27], were significantly increased in *B. breve* A1-administered control mice, but not in *B. breve* A1-administered KD mice, compared with saline-administered control mice (Figure 4A-a,b). By contrast, neuropsin mRNA levels were significantly decreased in saline-administered KD mice within 4 h after PTZ injection, whereas the administration of *B. breve* A1 prevented this decrease (Figure 4A-b). Unlike neuropsin, ILK mRNA levels in KD mice did not differ from those in the saline-administered control group regardless of whether *B. breve* A1 had been administered (Figure 4A-a). The mRNA expression levels of MMP-9, a zinc-dependent endopeptidase member that dynamically degrades and alters the ECM structure [28] and promotes neuronal plasticity through the cleavage of ECM proteins [19,20], showed an increasing tendency at 4 h after PTZ administration but immediately decreased at 24 h after PTZ administration (Figure 4A-c); however, no significant differences were found between the groups. On the other hand, the mRNA expression levels of Iba1 and BDNF were not significantly different after the administration of *B. breve* A1 in either KD or control mice; these levels only tended to decrease in KD mice within 4 h after PTZ injection, regardless of whether *B. breve* A1 had been administered (Figure 5A,B).

### 2.3. Effect of B. breve A1 on the Hippocampal Protein Expression Levels of ILK, Neuropsin, and MMP-9

Protein expression of ILK and neuropsin, the mRNA expression levels of which in control mice were increased by the administration of *B. breve* A1, were subjected to Western blot analysis. The expression level of ILK protein was significantly increased in only *B. breve* A1-administered control mice compared with saline-administered control mice (Figure 4B-a). However, the ILK protein expression levels in KD mice at 4 and 24 h after PTZ injection were the same as those in saline-administered control mice, regardless of whether *B. breve* A1 had been administered (Figure 4B-a). On the other hand, the expression level of neuropsin protein was significantly decreased in KD mice at only 4 h after PTZ injection; however, the administration of *B. breve* A1 prevented this decrease (Figure 4B-b). The administration of *B. breve* A1 significantly increased the expression level of neuropsin mRNA in control mice (Figure 4A-b) but did not change the level of neuropsin protein (Figure 4B-b). Conversely, the expression level of MMP-9 mRNA varied greatly over time after PTZ administration (Figure 4A-c), but the MMP-9 protein level showed an increasing tendency over time after PTZ administration and was significantly increased in only *B. breve* A1-administered KD mice at 24 h after PTZ injection compared with saline-administered control mice (Figure 4B-c). Appendix A summarizes the results of the mRNA and protein expression levels (Appendix A).

### 2.4. Effect of B. breve A1 on the Phosphorylation of Akt Ser^473^ (p-Akt) Expression in the Hippocampus

ILK is known to activate Akt, one of the kinases in the cell survival pathway, by phosphorylation at Serine 473 [29,30]. The protein expression level of p-Akt was significantly increased at 4 and 24 h after PTZ administration only in *B. breve* A1-treated KD mice (Figure 6). On the other hand, saline-treated KD mice showed an increasing tendency only at 24 h after PTZ administration, although this difference was not significant. Moreover, no significant differences in the p-Akt protein expression levels were seen in *B. breve* A1-administered control mice or saline-administered KD mice at 4 h after PTZ injection compared with saline-administered control mice.

### 2.5. Effect of the ILK Inhibitor Cpd22 on the Antiepileptic Effect of B. breve A1

To investigate whether the enzymatic activity of ILK is involved in the effect of *B. breve* A1 on PTZ-induced seizures, Cpd22 (10 mg/kg), a selective inhibitor of ILK enzyme activity [31], was administered intraperitoneally 60 min before the fourth PTZ injection on day 7 (Figure 1B). The administration of Cpd22 to *B. breve* A1-treated KD mice nullified the inhibitory effect of *B. breve* A1 on PTZ-induced seizures and significantly increased seizure scores in *B. breve* A1-treated KD mice (Figure 7). Moreover, the administration of Cpd22 to saline-treated KD mice stimulated PTZ-induced seizures, resulting in significantly increased seizure scores (Figure 7). These results suggest that the activity of ILK is directly involved in the *B. breve* A1-induced suppression of seizures induced by the repeated administration of PTZ and that the enzymatic activity of endogenous ILK might physiologically work in a direction to antagonize PTZ-induced seizures, even in mice not treated with *B. breve* A1.

### 2.6. Effect of the ILK Inhibitor Cpd22 on p-Akt Expression Levels in KD Mice with or without the Administration of B. breve A1

Cpd22 (10 mg/kg) was administered intraperitoneally 60 min before the fourth PTZ injection on day 7 (Figure 1B). The protein expression level of p-Akt was significantly increased at 4 h after PTZ injection in vehicle-treated *B. breve* A1-administered KD mice compared with vehicle-treated saline-administered KD mice (Figure 8). By contrast, treatment with Cpd22 reduced the increased p-Akt expression levels in the vehicle-treated *B. breve* A1-administered KD mice to the levels of the vehicle-treated saline-administered KD mice (Figure 8). However, p-Akt expression levels in saline-administered KD mice were not significantly different after treatment with Cpd22, maintaining the same levels with or without Cpd22. The same result was confirmed in naïve mice, i.e., Cpd22 had no effect on the basal expression level of p-Akt in the hippocampus of naïve mice (Appendix A). These results imply that basal p-Akt levels in saline-treated KD mice are maintained by an ILK-independent signaling pathway and that ILK enzyme activity is responsible for the *B. breve* A1-induced increase in p-Akt expression levels but not for the basal p-Akt levels. Nevertheless, Cpd22 stimulated the seizure scores in saline-treated KD mice (Figure 7). Therefore, ILK-mediated signaling molecules other than p-Akt might function in a direction to antagonize repeated PTZ-induced seizures.

## 3. Discussion

In the present study, we investigated the effects of oral administration of *B. breve* A1 on tonic–clonic seizure in PTZ-induced KD mice to evaluate its antiepileptic potential. The results indicate that the oral administration of *B. breve* A1 partially but significantly ameliorates tonic–clonic seizure in PTZ-induced KD mice; however, the mechanism underlying this amelioration is by virtue of still-unknown downstream effectors of the ILK-Akt signaling pathway. It has been reported that *B. breve* A1 prevents cognitive impairment in AD model mice [12] and patients with AD [13] and restores abnormal hippocampal synaptic plasticity and cognitive deficits in PD mice [11]. Using PTZ-induced kindling mice, we found for the first time that the oral administration of *B. breve* A1 has a new indication for antiepileptic activity.

The luminal gut microbiota is considered to communicate with the brain bidirectionally through multiple signaling pathways derived from the neuronal, endocrine, and immune systems, other than via the nutrients and by-products of microbial metabolism [32]. Moreover, perturbation of this microbiota–gut–brain axis has been considered to be involved in neurodegenerative disorders [33]. In the present study, the administration of nonviable *B. breve* A1 did not show any significant effects on mean seizure scores after PTZ injection in KD mice (Figure 3). Similarly, we previously reported that living but not nonviable *B. breve* A1 improved cognitive impairment in PD mice [11]. On the other hand, the administration of nonviable *B. breve* A1 to AD model mice, which was less effective than viable *B. breve* A1, partially ameliorated the cognitive impairment observed in AD model mice [12]. These results imply that the different signals derived from intra-gut living and/or nonviable *B. breve* A1 transmit to the central nervous system and exert beneficial effects on different neurological disease model mice.

Epileptogenicity is also considered to be associated with changes in synaptic plasticity [23,24,25]. Therefore, we examined the mRNA expression levels of several proteins involved in synaptic plasticity and reported to change in epilepsy animal models. The results indicated a significant decrease in the mRNA expression levels of neuropsin in KD mice at 4 h after PTZ injection; however, the administration of *B. breve* A1 prevented this decrease (Figure 4A-b). The same results were obtained by Western blotting analysis; the protein expression level of neuropsin was significantly decreased in KD mice at 4 h after PTZ injection compared with saline-administered control mice, whereas the administration of *B. breve* A1 recovered the decreased expression level to that of saline-administered control mice (Figure 4B-b). Therefore, *B. breve* A1 may restore the altered synaptic plasticity to normal. MMP-9 protein induction and the upregulation of its enzymatic activity at hippocampal synapses during epileptic seizures have been reported to be closely related to epileptogenicity [34]. In the present study, the mRNA expression levels of MMP-9 showed no significant changes between groups, but there was an increasing tendency at 4 h after PTZ administration, followed by an immediate decrease at 24 h after PTZ administration (Figure 4A-c). Thus, the mRNA induction of MMP-9 may be transiently stimulated after PTZ-induced seizure in KD mice. However, in spite of the MMP-9 mRNA level, the MMP-9 protein level showed an increasing tendency over time after PTZ administration and was significantly increased in only *B. breve* A1-administered KD mice at 24 h after PTZ injection compared with saline-administered control mice (Figure 4B-c). The reason for this discrepancy in the MMP-9 mRNA and protein expression levels over time after PTZ administration remains unclear. On the other hand, BDNF is known to have a protective role against excitotoxicity-mediated PTZ-induced seizure [35], and microglia have been shown to be activated following PTZ-induced chronic epilepsy [36]. However, no significant changes in the mRNA expression of BDNF or Iba1 were observed in KD mice, regardless of the administration of saline or *B. breve* A1 (Figure 5). These results suggest that BDNF and microglia might not be involved in the antiepileptic activity of *B. breve* A1.

The results of reverse transcription–quantitative polymerase chain reaction (RT–qPCR) and Western blot analyses revealed that both the mRNA and protein expression of ILK were enhanced in only *B. breve* A1-administered control mice (Figure 4A-a,B-a), suggesting that *B. breve* A1 stimulates the induction of ILK protein. In contrast to *B. breve* A1-administered control mice, no significant changes in ILK mRNA or protein levels were observed in *B. breve* A1-administered KD mice. The reason for this remains unknown, but neuronal damage after PTZ-induced seizure might affect mRNA and protein induction in KD mice. It has been reported that ILK and its downstream molecule Akt in the integrin-mediated cell survival signaling pathway exert neuroprotective effects against hippocampal cell death after epileptic seizure [21,37]. Therefore, we examined the protein expression level of p-Akt after PTZ injection. The expression level of p-Akt was significantly increased at 4 and 24 h after PTZ injection in only *B. breve* A1-treated KD mice (Figure 6). By contrast, saline-treated KD mice showed no significant changes in p-Akt levels but an increasing tendency at 24 h after PTZ administration. These results suggest that Akt activation after PTZ injection is further stimulated by the administration of *B. breve* A1, resulting in the attenuation of PTZ-induced seizure in KD mice.

ILK is a cytoplasmic serine/threonine kinase that mediates signal transduction derived from integrin and growth factors [38]. ILK interacts with the cytoplasmic domain of β1-integrin and functions as a scaffold protein bridging integrin and the actin cytoskeleton and various intracellular signaling pathways that regulate cell survival, proliferation, and migration [39]. ILK is also known to activate Akt, one of the kinases in the cell survival pathway [29,30], by phosphorylation at Serine 473. To examine directly whether ILK is involved in Akt activation and the subsequent inhibition of seizures, we examined the effect of Cpd22 on the stimulated p-Akt expression levels and antiepileptic action in *B. breve* A1-administered KD mice. Cpd22 completely reduced the increased p-Akt expression levels of *B. breve* A1-treated KD mice without affecting p-Akt levels in saline-treated KD mice (Figure 8). At the same time, Cpd22 nullified the inhibitory effect of *B. breve* A1 on PTZ-induced seizures and increased the seizure score (Figure 7). However, Cpd22 significantly increased seizure scores in not only *B. breve* A1-treated but also saline-treated KD mice. These results suggest that ILK is involved in the *B. breve* A1-induced stimulation of Akt activation and the antiepileptic action of *B. breve* A1. However, because Cpd22 significantly increased seizure scores in the saline-treated KD mice, the p-Akt expression levels of which were not significantly altered with or without the administration of Cpd22 (Figure 8), it seems that ILK-mediated signaling molecules other than p-Akt might function in a direction to antagonize repeated PTZ-induced seizures.

Regarding the role of phosphoinositide 3-kinase (PI3K)-Akt signaling in epileptic seizures, a protective effect on neuronal impairment [37,40,41] has been reported. ILK activity is also regulated in a PI3K-dependent manner, and activated ILK directly phosphorylates Akt at Serine 473, resulting in its activation [29]. In the present study, the administration of *B. breve* A1 in KD mice caused the activation of ILK-Akt signaling, leading to the stimulation of p-Akt expression levels and the amelioration of PTZ-induced seizures. As *B. breve* A1 might attenuate seizure-induced hippocampal neuronal damage via activation of ILK-Akt signaling, this neuroprotective action may reduce seizure susceptibility in KD mice following chronic administration of PTZ (Figure 9).

MMPs have been reported to be implicated in the neuropathology of various diseases other than in the physiological functions of the brain [43,44,45]. It has been reported that MMP-9 protein induction and the upregulation of its enzymatic activity at hippocampal synapses during epileptic seizures are closely related to epileptogenicity [34]. In contrast, the antiepileptic drug diazepam has been shown to inhibit the development of PTZ-induced kindled seizures and increased MMP-9 levels in the hippocampus [18]. Moreover, a selective MMP-9 inhibitor and the polymethoxylated flavonoid tangeretin, which has an antiepileptic effect via MMP-9 downregulation, have been reported to prevent seizure-induced hippocampal damage via the activation of ILK and PI3K-Akt signaling [21,36]. Kim et al. [21] suggested that hippocampal neuronal cell death after epileptic seizure occurs when MMP-9-mediated extracellular matrix proteolysis causes the downregulation of β1-integrin, resulting in the dysfunction of integrin-mediated cell survival signaling, including the ILK and PI3K-Akt-mediated signaling pathways.

In the present study, although MMP-9 mRNA levels showed a transient increasing tendency at 4 h after PTZ administration, they immediately decreased at 24 h after PTZ administration (Figure 4A-c). On the contrary, MMP-9 protein levels were significantly increased in only *B. breve* A1-administered KD mice at 24 h after PTZ injection compared with saline-administered control mice (Figure 4B-c). As the interpretation of these results remains inconclusive, it is important to gain a better understanding of the relationship between MMP-9 enzyme activity and the antiepileptic effect of *B. breve* A1.

Overall, our results indicate that the oral administration of *B. breve* A1 ameliorates tonic–clonic seizure in KD mice via ILK-mediated Akt activation. Therefore, ILK-Akt signaling, one of the signals in the cell survival pathway, might attenuate seizure-induced hippocampal neuronal damage, reduce seizure susceptibility, and restore altered synaptic plasticity in KD mice following the chronic administration of PTZ (Figure 9). However, the mechanisms underlying the activation of *B. breve* A1-induced ILK-Akt signaling and its downstream effectors remain unknown. Further studies are needed to investigate whether *B. breve* A1 is as effective as PTZ-induced kindling in the seizures of chronic epilepsy model mice treated with pilocarpine or kainic acid and also to clarify the mechanisms underlying the activation of *B. breve* A1-induced ILK-Akt signaling, its downstream effectors, and the signaling pathway of living *B. breve* A1 via the microbiota–gut–brain axis.

## 4. Materials and Methods

### 4.1. Animals and PTZ Treatment

Male C57BL/6 mice (age range: 7–8 weeks) (Ishibe Breeding Facility, Clea Japan, Tokyo, Japan) were maintained under controlled temperature (22 ± 2 °C) and humidity (35 ± 5%) on a 12 h light/12 h dark cycle (lights on at 07:00) and allowed ad libitum access to pellet food (Clea Japan) and water. All procedures for the care and use of experimental animals were approved by the Animal Research Committee at Obihiro University of Agriculture and Veterinary Medicine (No. 22-1 and No. 23-4) and conducted in compliance with the 1989 Guiding Principles for the Use of Animals in Toxicology.

PTZ-induced KD mice were prepared following Shimada and Yamagata [15]. PTZ (Sigma-Aldrich Japan, Tokyo, Japan) was dissolved in saline to prepare concentrations of 3.1 mg/mL. PTZ (37 mg/kg) was administered to the mice intraperitoneally every other day for 15 days (Figure 1A). Saline-administered control mice (Control-Saline) were administered saline instead of PTZ intraperitoneally every other day for 15 days. In the experiment to test the effect of the ILK inhibitor Cpd22 on the antiepileptic effect of *B. breve* A1, PTZ (37 mg/kg) was administered intraperitoneally to mice every other day for 7 days so that the mice would not die from intensified seizures (Figure 1B). Saline was administered in a similar manner as a control. Seizure scores gradually increased with PTZ injections, whereas no seizures were observed in the control mice. The behavior of the mice was observed for 30 min after each PTZ injection. The resultant seizures were scored as follows [15]: stage 0, normal behavior, no abnormality; stage 1, immobilization, lying on belly; stage 2, head nodding, facial, forelimb, or hindlimb myoclonus; stage 3, continuous whole-body myoclonus, myoclonic jerks, tail held up stiffly; stage 4, rearing, tonic seizure, falling down on the side; stage 5, tonic–clonic seizure, falling down on the back, wild rushing and jumping; and stage 6, death. The dose and number of administrations of PTZ were determined by observing the behavior of the PTZ-injected mice that did not exceed the stage 5 seizure score.

### 4.2. RNA Extraction and RT–qPCR Assay

Total RNA was extracted from hippocampal tissue using Direct-zol™ RNA MiniPrep (Zymo Research, Tustin, CA, USA) and quantified using the QuantiFluor™ RNA system (Promega, Madison, WI, USA) according to the manufacturers’ instructions. RNA was amplified using the MyGo Pro Real-Time PCR system (IT-IS Life Science, Ltd., Cork, Ireland). One-step RT–qPCR was performed using MyGo Green 1-step Low Rox (IT-IS Life Science, Ltd.) for a total volume of 20 μL and a template concentration of 10 pg/μL total RNA, according to the manufacturer’s recommendations. The thermal cycling conditions were 45 °C for 10 min as an RT step and 95 °C for 2 min, followed by 40 cycles of 95 °C for 10 s and 60 °C for 20 s. The relative quantification (fold change) of mRNA expression was estimated by the use of the 2^−ΔΔCt^ method [46], as described in our previous report [11]. β-Actin was used as the housekeeping gene for each sample to normalize the targeted gene expression. The primers used in this study are shown in Table 1.

### 4.3. Western Blot Analysis

The mice were anesthetized with isoflurane and then decapitated. The brains were removed and transferred to ice-cold PBS, and then the hippocampi were rapidly dissected from each brain on ice. Hippocampal tissue was solubilized in three volumes of Cell lysis buffer (Cell Signaling Technology, Tokyo, Japan) with 0.25% sodium deoxycholate, to which 1 mM phenylmethylsulfonyl fluoride (PMSF) was added immediately prior to use, at 4 °C and then centrifuged at 14,000× *g* for 10 min. The supernatant was collected, and the protein concentration was determined using the BCA Protein Assay Kit (Takara Bio, Shiga, Japan). After mixing with an equal volume of 2 × sodium dodecyl sulphate (SDS)-sample buffer (2% SDS, 10% glycerol, 10% 2-mercaptoethanol, 0.1% bromophenol blue, and 62.5 mM Tris-HCl, pH 6.8) and then filtrating through Ultrafree-MC filter devices (5.0 μm, MilliporeSigma, Burlington, MA, USA), equal amounts of protein were subjected to polyacrylamide gel electrophoresis (10% polyacrylamide) and transferred onto nitrocellulose membranes at 4 °C in 25 mM Tris-HCl, 192 mM glycine, 20% methanol, and 0.025% SDS. After blocking with 4% Block Ace (KAC, Tokyo, Japan) and dissolving in PBS containing 0.1% (*v*/*v*) Tween 20 (TBST) for 1 h at room temperature, the membranes were incubated with rabbit anti-neuropsin polyclonal antibody [ab232839] (abcam, Tokyo, Japan; 1:500 in TBST), mouse anti-MMP-9 monoclonal antibody (abcam; 1:500 in TBST), rabbit anti-ILK polyclonal antibody (Cell Signaling Technology, Danvers, MA, USA; 1:1000 in TBST), rabbit anti-phospho-Akt polyclonal antibody (Cell Signaling Technology; 1:1000 in TBST), rabbit anti-Akt polyclonal antibody (Cell Signaling Technology; 1:1000 in TBST), rabbit anti-β-tubulin polyclonal antibody (Proteintech, Tokyo, Japan; 1:3500 in TBST), or mouse anti-glyceraldehyde-3-phosphate dehydrogenase (GAPDH) monoclonal antibody (Proteintech; 1:3000 in TBST) at 4 °C overnight. After incubation, the membranes were washed three times in TBST and probed with goat anti-mouse IgG antibody or goat anti-rabbit IgG conjugated to horseradish peroxidase (Cell Signaling Technology; 1:3000 in TBST) at room temperature for 1 h. After washing the membranes three times in TBST, the final protein-IgG complexes were identified using Amersham ECL Western Blotting Detection reagents (Cytiva, Tokyo, Japan), followed by detection with LAS3000 (Fuji Photo Film, Tokyo, Japan). Assessment of the band intensities was performed using Multi-Gauge software ver. 3 (Fuji Photo Film). The protein levels were normalized to β-tubulin or GAPDH as a loading control.

### 4.4. Administration of B. breve A1

*B. breve* A1, a probiotic strain stocked as strain MCC1274, was derived from commercial products (Morinaga Milk Industry Co., Ltd., Tokyo, Japan). Lyophilized *B. breve* A1 was suspended in saline at a concentration of 1.3 × 10^10^ cfu/mL just before administration. Nonviable *B. breve* A1 was prepared by further heat-shock treatment of the suspended bacterium at 60 °C for 60 min and stored at –20 °C until use, following a previous report [12]. Mice were orally administered living or heat-killed *B. breve* A1 at a volume of 0.25 mL (3.25 × 10^9^ cfu organisms) every other day for 15 days, the administration of which started on the day before the first PTZ injection (Figure 1). In the experiment to test the effect of the ILK inhibitor Cpd22 on the antiepileptic effect of *B. breve* A1, the mice were orally administered *B. breve* A1 every other day for 7 days (Figure 1B). For oral gavage, a mouse feeding stainless steel bulbous-ended needle (0.92 × 70 mm; AS ONE, Osaka, Japan) inserted over the tongue into the stomach was used.

### 4.5. Administration of Cpd22

Mice were given an intraperitoneal injection of Cpd22 (Merck Millipore, Tokyo, Japan) at a dose of 10 mg/kg in saline containing 1% dimethyl sulfoxide (DMSO) or the vehicle 60 min prior to the fourth PTZ injection (Figure 1B).

### 4.6. Data Analysis

According to the analyses of homoscedasticity using Levene’s test, multiple group comparisons were assessed using one-way analysis of variance followed by Tukey’s post hoc test or by Kruskal–Wallis and Mann–Whitney tests with Bonferroni corrections. Statistical differences were considered significant when *p* < 0.05. All statistical analyses were performed using SPSS software (version 16.0; SPSS Japan, Inc., Tokyo, Japan).

## 5. Conclusions

We found that the oral administration of *B. breve* A1 to KD mice ameliorated PTZ-induced seizures through the activation of the ILK-Akt signaling pathway. The antiepileptic effect of *B. breve* A1 was observed when living but not nonviable *B. breve* A1 was administered to KD mice. Significant decreases in the mRNA and protein expression levels of neuropsin were observed in KD mice at 4 h after PTZ injection, but the administration of *B. breve* A1 significantly recovered these to control levels. Moreover, *B. breve* A1-induced Akt activation occurred at 4 and 24 h after PTZ injection, which was blocked completely by the ILK inhibitor Cpd22. At the same time, Cpd22 nullified the inhibitory effect of *B. breve* A1 on PTZ-induced seizures. Therefore, the mechanisms underlying the antiepileptic effect of *B. breve* A1 are considered to be via still-unknown downstream effectors of the ILK-Akt signaling pathway. These findings suggest that *B. breve* A1 could be useful as a therapeutic probiotic in the treatment of epilepsy in combination with antiepileptic drugs.

## 6. Patents

Patent Application in Japan, No. 2022-089546: Antiepileptic compositions [Inventor: Dr. Toshiaki Ishii].

## Figures and Tables

**Figure 1 ijms-25-09259-f001:**
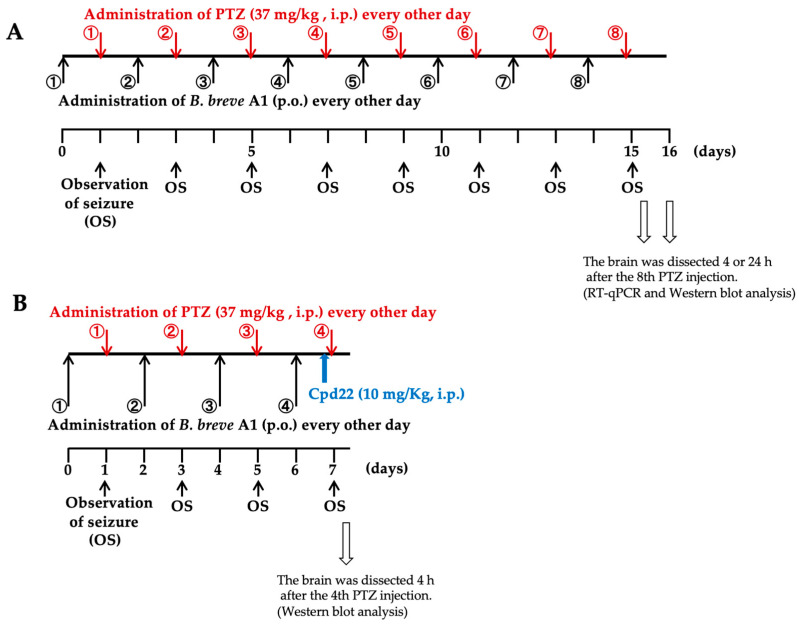
A schematic figure explaining the planning of the experiments. (**A**) Eight-week-old mice were given intraperitoneal injections of PTZ at a single dose of 37 mg/kg every other day for 15 days. (**B**) In the experiment to test the effect of the ILK inhibitor Cpd22 on the antiepileptic effect of *B. breve* A1, PTZ (37 mg/kg) was administered intraperitoneally to mice every other day for 7 days so that they would not die from intensified seizures. The numbers in circles indicate the number of administrations of PTZ (red) and *B. breve* A1 (black) to explain how many administrations have been made since the first administration (①).

**Figure 2 ijms-25-09259-f002:**
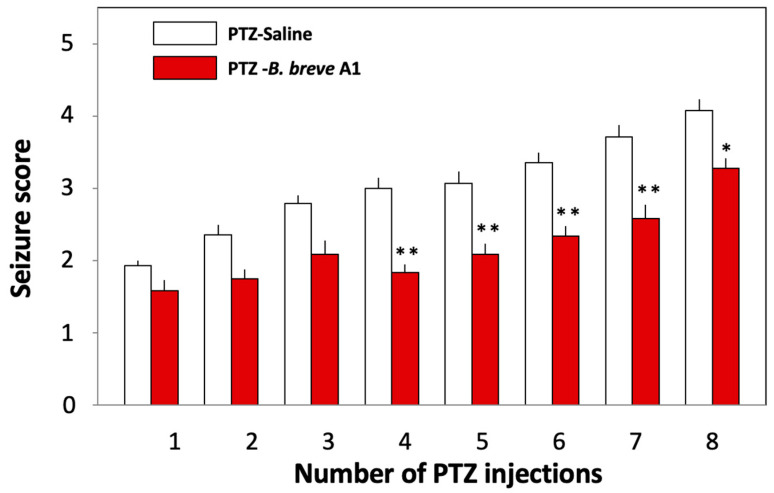
Effects of *B. breve* A1 on PTZ-induced seizures in KD mice. Mice were orally administered living *B. breve* A1 at a volume of 0.25 mL (3.25 × 10^9^ cfu organisms suspended in saline) or saline every other day for 15 days, the administration of which started on the day before the first PTZ injection (Figure 1A). Data are expressed as the mean ± SEM; *n* = 12–14. * *p* < 0.05 and ** *p* < 0.01 between PTZ-Saline and PTZ-*B. breve* A1 on the same number of PTZ injections (one-way analysis of variance (ANOVA) followed by Tukey’s post hoc test).

**Figure 3 ijms-25-09259-f003:**
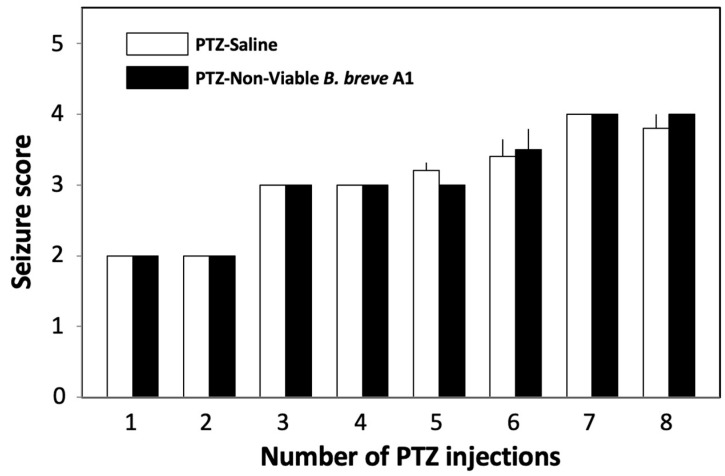
Effects of nonviable *B. breve* A1 on PTZ-induced seizures in KD mice. Mice were orally administered nonviable *B. breve* A1 at a volume of 0.25 mL (3.25 × 10^9^ cfu organisms suspended in saline) or saline every other day for 15 days, the administration of which started on the day before the first PTZ injection (Figure 1A). The behavior of the mice was observed for 30 min after each PTZ injection. The resultant seizures were scored. Data are expressed as the mean ± SEM; *n* = 4–5. No significant differences were observed between groups (one-way ANOVA followed by Tukey’s post hoc test).

**Figure 4 ijms-25-09259-f004:**
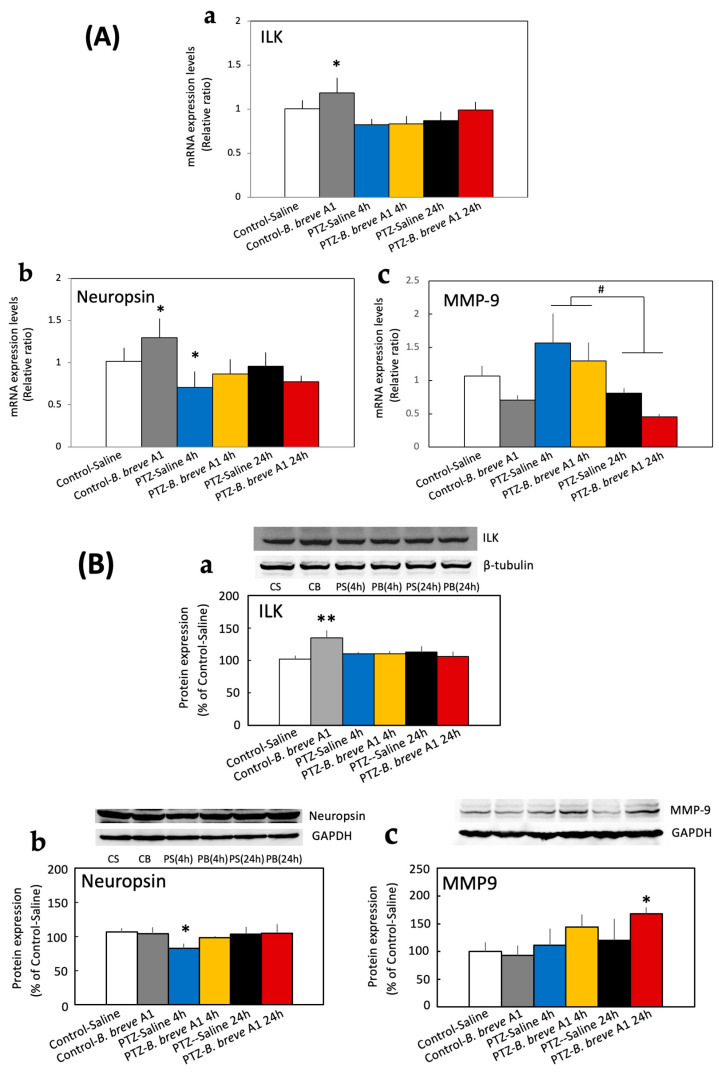
(**A**) RT–qPCR analysis of the hippocampal mRNA expression levels of ILK (**a**), neuropsin (**b**), and MMP-9 (**c**) in saline-administered controls and KD mice at 4 and 24 h after PTZ injection, and in *B. breve* A1-administered controls and KD mice at 4 and 24 h after PTZ injection. Data are expressed as the mean ± SD: *n* = 6–9 per group. * *p* < 0.05 vs. Control-Saline. # *p* = 0.004 between 4 h and 24 h after PTZ injection (one-way ANOVA followed by Tukey’s post hoc test). (**B**) ILK (**a**), neuropsin (**b**), and MMP-9 (**c**) protein expression levels in saline-administered controls and KD mice at 4 and 24 h after PTZ injection, and in *B. breve* A1-administered controls and KD mice at 4 and 24 h after PTZ injection. The images show representative results [(**a**) upper: ILK, lower: β-tubulin, (**b**) upper: neuropsin, lower: GAPDH, (**c**) upper: MMP-9, lower: GAPDH]. The protein levels were normalized to β-tubulin as a loading control. The results are shown as a percentage of protein expression levels in Control-Saline. Data are expressed as the mean ± SD: *n* = 3–4 per group. * *p* < 0.05 and ** *p* < 0.01 vs. Control-Saline (one-way ANOVA followed by Tukey’s post hoc test).

**Figure 5 ijms-25-09259-f005:**
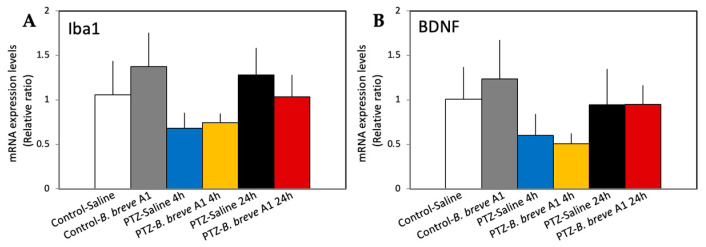
RT–qPCR analysis of the hippocampal mRNA expression levels of Iba1 (**A**) and BDNF (**B**) in saline-administered control and KD mice at 4 and 24 h after PTZ injection, and in *B. breve* A1-administered controls and KD mice at 4 and 24 h after PTZ injection. Data are expressed as the mean ± SD: *n* = 6–9 per group. No significant differences were observed between groups (Iba1: one-way ANOVA followed by the Kruskal–Wallis and Mann–Whitney tests with Bonferroni corrections, BDNF: one-way ANOVA followed by Tukey’s post hoc test).

**Figure 6 ijms-25-09259-f006:**
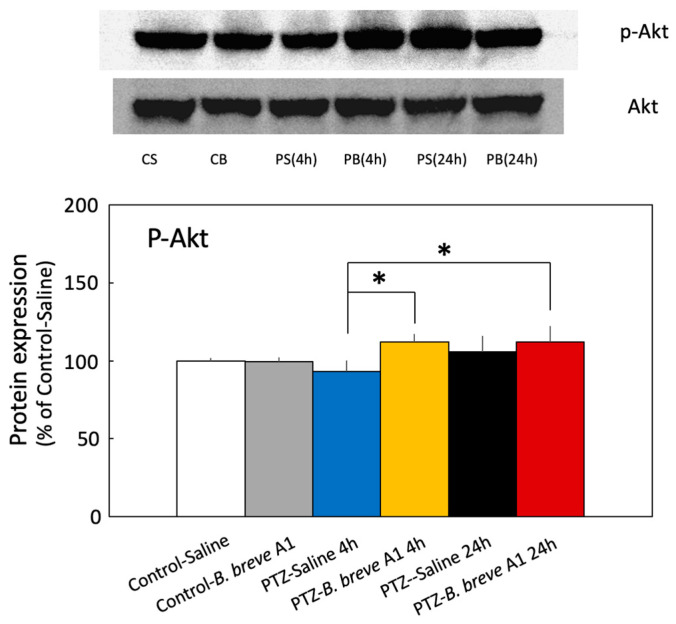
The administration of *B. breve* A1 to KD mice increased the phosphorylation of Akt Ser^473^ (p-Akt) expression levels in the hippocampus at 4 and 24 h after PTZ injection. The images show representative results (upper: p-Akt, lower: Akt). The p-Akt protein levels were normalized to Akt as a loading control. The results are shown as a percentage of protein expression levels in Control-Saline. Data are expressed as the mean ± SD: *n* = 4 per group. * *p* < 0.05 vs. PTZ-Saline 4 h (one-way ANOVA followed by Tukey’s post hoc test).

**Figure 7 ijms-25-09259-f007:**
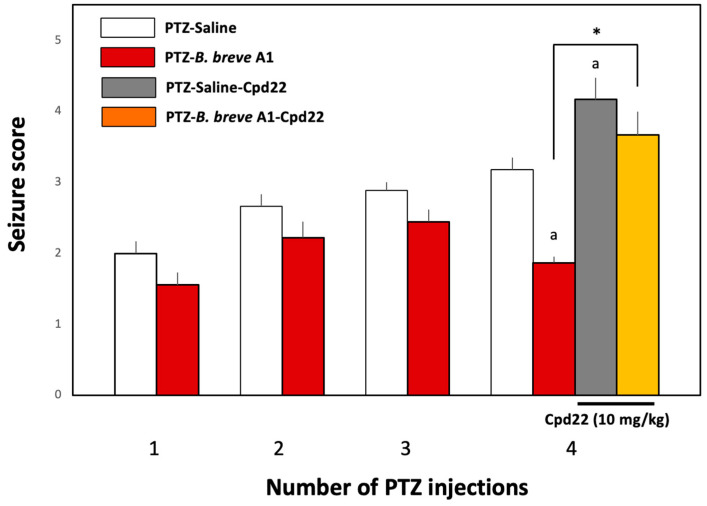
Effect of the ILK inhibitor Cpd22 on the antiepileptic effect of *B. breve* A1. Cpd22 (10 mg/kg) or the vehicle was administered intraperitoneally 60 min before the fourth PTZ injection on day 7 (Figure 1B). The behavior of the mice was observed for 30 min after each PTZ injection. The resultant seizures were scored. Data are expressed as the mean ± SEM; *n* = 6–12. * *p* < 0.01 between PTZ-*B. breve* A1 (the fourth PTZ injection) and PTZ-*B. breve* A1-Cpd22, ^a^
*p* < 0.01 vs. PTZ-Saline (the fourth PTZ injection) (one-way ANOVA followed by Tukey’s post hoc test).

**Figure 8 ijms-25-09259-f008:**
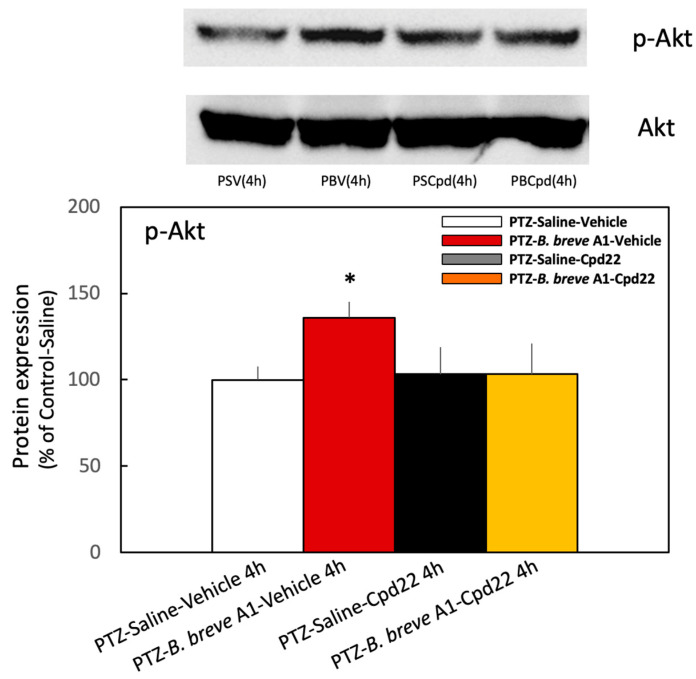
Effect of the ILK inhibitor Cpd22 on p-Akt expression levels in the hippocampus in saline-treated and *B. breve* A1-treated KD mice. Cpd22 (10 mg/kg) or the vehicle was administered intraperitoneally 60 min before the fourth PTZ injection on day 7, and the brain was dissected 4 h after PTZ injection (Figure 1B). The images show representative results (upper: p-Akt, lower: Akt). The p-Akt protein levels were normalized to Akt as a loading control. The results are shown as a percentage of protein expression levels in vehicle-treated saline-administered KD mice. Data are expressed as the mean ± SD: *n* = 3 per group. * *p* < 0.05 vs. PTZ-Saline-vehicle at 4 h (one-way ANOVA followed by Tukey’s post hoc test).

**Figure 9 ijms-25-09259-f009:**
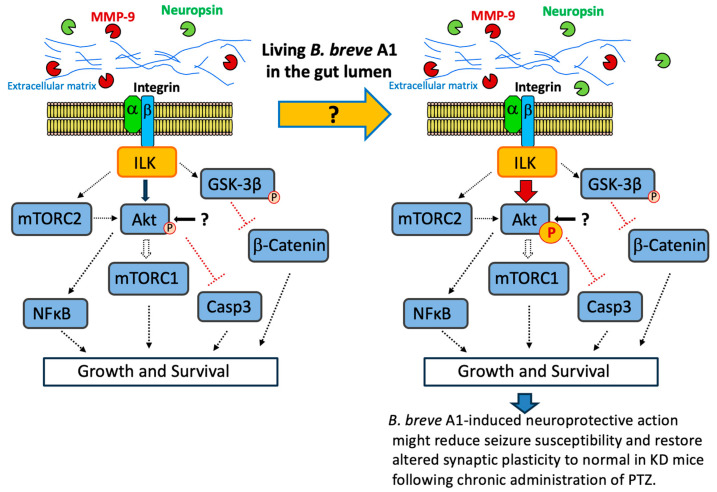
Living *B. breve* A1 in the gut lumen leads to a decrease in seizure susceptibility via an ILK-dependent signaling pathway, the mechanism of which is caused by a still-unknown signal via the gut–brain axis. ILK is a cytoplasmic serine/threonine kinase that mediates signal transduction derived from integrin and growth factors [38]. ILK interacts with the cytoplasmic domain of β1-integrin and functions as a scaffold protein bridging integrin and the actin cytoskeleton and various intracellular signaling pathways that regulate cell survival, proliferation, and migration [39]. In integrin-mediated cell survival signaling, ILK, one of the kinases in the cell survival pathway [29,30], and its downstream molecule Akt exert neuroprotective effects against hippocampal cell death after epileptic seizure [21,37]. The figure shows representative ILK-dependent cell growth and survival signaling pathways [42]. In the present study, *B. breve* A1-induced Akt activation occurred after PTZ injection, and the ILK inhibitor Cpd22 completely blocked its activation. At the same time, Cpd22 nullified the inhibitory effect of *B. breve* A1 on PTZ-induced seizures. Moreover, *B. breve* A1 recovered the decreased protein expression level of neuropsin, which is one of the plasticity-related extracellular proteases [26]. Therefore, we propose that *B. breve* A1-induced ILK activation promotes the phosphorylation of the key target signal proteins, including Akt, essential for integrin-dependent cell survival and proliferation, and exerts neuroprotective action against hippocampal cell death and abnormal changes in synaptic plasticity after epileptic seizures, resulting in decreased seizure susceptibility following the chronic administration of PTZ. Abbreviations: mTORC, mammalian target of rapamycin complex; Casp3, caspase-3; MMP, matrix metalloproteinase; GSK-3β, glycogen synthase kinase-3β.

**Table 1 ijms-25-09259-t001:** RT-qPCR primers in 5′–3′ direction.

Transcript	Primers	
β-Actin	Sense	ATTGCTGACAGGATGCAGAAG
	Antisense	TAGAAGCACTTGCGGTGCACG
Neuropsin	Sense	CTCAACTGTGCGGAAGTGAA
	Antisense	ACTCCAGGTTTCTCGGGTTT
ILK	Sense	AAGGTGCTGAAGGTTCGAGA
	Antisense	CAGTGTGTGATGAGGGTTGG
BDNF	Sense	GCGGCAGATAAAAAGACTGC
	Antisense	CTTATGAATCGCCAGCCAAT
Iba-1	Sense	GAAGCGAATGCTGGAGAAAC
	Antisense	GACCAGTTGGCCTCTTGTGT
MMP-9	Sense	CGTGTCTGGAGATTCGACTTGA
	Antisense	TGGAAGATGTCGTGTGAGTTCC

## Data Availability

The data presented in this study are available on request from the corresponding author.

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
