# Peer review of "Oral Administration of Probiotic Bifidobacterium breve Ameliorates Tonic–Clonic Seizure in a Pentylenetetrazole-Induced Kindling Mouse Model via Integrin-Linked Kinase Signaling"

_ijms, 2024, doi:10.3390/ijms25179259_

Round 1

Reviewer 1 Report

Comments and Suggestions for Authors

The authors identified a pathology with an important social impact, epilepsy, for which they tried to identify signaling pathways, using an animal model. This way of identifying multiple signaling pathways is important, especially if the effects of the intestinal microbiota on the central nervous system are studied.

The study proposed the use of Bifidobacterium breve, in different experimental and control conditions, to identify sources of errors and experimental limits. For additional certainty in the identification of signaling pathways, the evaluations were on different levels, from epilepsy scores to protein expression and protein quantity.

The authors also highlighted the spectacular statistical results (for MMP9, for example) and those that had no statistical significance, allowing the identification of signaling pathways. The authors highlighted the possible interpretations of the results, with rigorous scientific caution.

For increased scientific safety, the authors made experimental batches including Cpd22 to quantify the effect of Akt kinases, as a simplifying model of neuronal survival.

The results are significant and require the encouragement of the continuation of this effort in order to bring new therapeutic opportunities.

Author Response

Comments 1: The results are significant and require the encouragement of the continuation of this effort in order to bring new therapeutic opportunities.

Response: We greatly appreciate your comments. Toward the development of probiotics that can contribute to the treatment of epilepsy, we will continue to investigate the effects of B. breve A1 on seizures in other mouse models of chronic epilepsy, and hope to elucidate the mechanisms underlying the activation of B. breve A1-induced ILK-Akt-signaling, its downstream effectors, and the signaling pathway of living B. breve A1 via the microbiota–gut–brain axis.

Reviewer 2 Report

Comments and Suggestions for Authors

The aim of this study is to assess the effectiveness of a probiotic in reducing seizure scores in epileptic mice. 

Epilepsy is a chronic neurological disorder characterized by recurrent seizures that affects over 70 million people worldwide. The authors found that oral administration of Bifidobacterium breve strain A1 can have anticonvulsant effects. Using a PTZ kindled mouse model, the authors discovered that B. breve A1 administration reduced seizure scores and increased the level of Akt phosphorylation in the hippocampus, indicating potential antiepileptic effects. 

This study presents a new opportunity to explore the potential of probiotics as an anti-convulsant.

I'm interested in finding out if this treatment method would be effective for chronic epilepsy models of pilocarpine and kainic acid. 

I think the authors should conduct an extra control experiment to investigate the effect of Cpd22 in naïve mice in order to observe the baseline effects of this inhibitor. 

Additionally, I recommend that the authors combine figures 4 and 6.

Conclusion is consistent; however, the study's scope is limited as all the data has been collected from a PTZ model.

Comments on the Quality of English Language

The manuscript contains numerous grammatical errors.

Author Response

Comments 1: I'm interested in finding out if this treatment method would be effective for chronic epilepsy models of pilocarpine and kainic acid.

Response: Thank you for your comment. We too are interested in the effects of B. breve A1 on seizures in other mouse models of chronic epilepsy. Therefore, we plan to investigate whether B. breve A1 is as effective as PTZ-induced kindling in the seizures of chronic epilepsy model mice treated with pilocarpine or kainic acid. In response, we have added the sentences about this issue in the section of Discussion (page 12, line 372). Page 12, line 372 now read “Further studies are needed to investigate whether B. breve A1 is as effective as PTZ-induced kindling in the seizures of chronic epilepsy model mice treated with pilocarpine or kainic acid,…”

Comments 2: I think the authors should conduct an extra control experiment to investigate the effect of Cpd22 in naïve mice in order to observe the baseline effects of this inhibitor.

Response: Thank you for your comments. We agree with your idea to show the baseline effects of Cpd22 in naïve mice. In response, we have added a new supplemental figure (Figure S1) showing the results of the effect of Cpd22 on the basal expression level of p-Akt in the hippocampus of naïve mice. Also, we have revised and added the sentence about this issue in the section of Result (page 8, line 214). Page 8, line 214 now read “The same result was confirmed in naïve mice, i.e., Cpd22 had no effect on the basal expression level of p-Akt in the hippocampus of naïve mice (Figure S1).”

Comments 3: Additionally, I recommend that the authors combine figures 4 and 6.

Response: Thank you for your comments. In response, as you recommend, Figure 4 and Figure 6 have been combined. In the revised manuscript, the original Figure 4 is shown as Figure 4A and the original Figure 6 is shown as Figure 4B, with changes to each.

Comments 4: Conclusion is consistent; however, the study’s scope is limited as all the data has been collected from a PTZ model.

Response: Thank you for your comments. We agree with you. As mentioned in the response to Comments 1, we would like to investigate the antiepileptic effect of B. breve A1 using other epilepsy model mice induced by pilocarpine or kainic acid in the future. We believe that it will be important to clarify the mechanism of action of the antiepileptic effect of B. breve A1.

Reviewer 3 Report

Comments and Suggestions for Authors

The study explores the effects of the probiotic Bifidobacterium breve A1 on seizure activity in a mouse model. The research indicates that oral administration of B. breve A1 reduces seizure scores and suggests that the mechanism involves integrin-linked kinase (ILK) signaling in the hippocampus, particularly influencing Akt phosphorylation which plays a role in neuroprotection and seizure resistance.

The conclusions drawn from the study are supported by the results, which show that B. breve A1 can significantly reduce seizure scores and influence key molecular pathways associated with epilepsy.

1) The methods are generally well-described, but additional details on the controls used and the statistical methods for data analysis would enhance reproducibility and clarity.

2) The results are informative; however, the presentation could be enhanced by organizing the data into more detailed tables and figures that summarize the findings more succinctly, particularly the data on protein and mRNA expression levels.

3) While the introduction covers the basic epidemiology and current treatments of epilepsy, it could benefit from a more detailed review of previous studies specifically focusing on the gut-brain axis and probiotic interventions in neurological disorders. Riva and colleagues recently showed that the ratio of two dominant phyla (Bacteroidota-to-Firmicutes) was similarly increased in Epi and No-Epi rats vs sham control rats. Notably, the relative abundance of families, genera, and species associated with SCFA production differed in Epi vs No-Epi rats, describing a bacterial imprint associated with epilepsy. 

Comments on the Quality of English Language

 The manuscript is generally well-written, but there are occasional grammatical errors and awkward phrasings that could be refined to improve readability and professionalism.

Author Response

Comments 1: The methods are generally well-described, but additional details on the controls used and the statistical methods for data analysis would enhance reproducibility and clarity.

Response: Thank you for your comments. In response, we have specified the statistical methods for data analysis in each figure legend. Page 3, line 101 (in Figure 2. Legend), page 4, line 108 (in Figure 3. Legend), page 6, line 155 and line 162 (in Figure 4(A) and 4(B) Legend), page 7, line 183 (in Figure 6. Legend), page 8, line 203 (in Figure 7. Legend), and page 9, line 229 (in Figure 8. Legend), now read “(one-way analysis of variance (ANOVA) followed by Tukey’s post-hoc test).” Also, page 6, line 167 (in Figure 5. Legend) now read “(Iba1: one-way ANOVA followed by Kruskal-Wallis and Mann-Whitney tests with Bonferroni corrections, BDNF: one-way ANOVA followed by Tukey’s post-hoc test)” In addition, we have added a sentence to describe saline-administered control mice in the section of Materials and Methods (page 12, line 389). Page 12, line 389 now read “Saline-administered control mice (Control-Saline) was administered saline instead of PTZ intraperitoneally every other dayfor 15 days.”  

Comments 2: The results are informative; however, the presentation could be enhanced by organizing the data into more detailed tables and figures that summarize the findings more succinctly, particularly the data on protein and mRNA expression levels.

Response: Thank you for your comments. In response, we have added a new supplemental table (Table S1.) summarizing the results of mRNA and protein expression levels. We also revised and added the sentence about this issue in the section of Results (page 5, line 148). Page 5, line 148 now read “Table S1 summarizes the results of mRNA and protein expression levels (Supplementary File).”

Comments 3:  While the introduction covers the basic epidemiology and current treatments of epilepsy, it could benefit from a more detailed review of previous studies specifically focusing on the gut-brain axis and probiotic interventions in neurological disorders. Riva and colleagues recently showed that the ratio of two dominant phyla (Bacteroidota-to-Firmicutes) was similarly increased in Epi and No-Epi rats vs sham control rats. Notably, the relative abundance of families, genera, and species associated with SCFA production differed in Epi vs No-Epi rats, describing a bacterial imprint associated with epilepsy. 

Response: We greatly appreciate your comments. In response, we have newly added the following report, Riva, A. et al., Neurobiology of Disease2024, 194, 106469. as the number of [10] in the list of References, and also revised and added the sentence about this issue in the section of Introduction (page 2, line 45). Page 2, line 45 now read “Moreover, an epilepsy-linked gut microbiota signature in a pediatric rat model of acquired epilepsy has been reported [10].”  

Round 2

Reviewer 3 Report

Comments and Suggestions for Authors

The revised paper has been improved. Thanks